# Gut Microbiota Dynamics in Natural Populations of *Pintomyia evans**i* under Experimental Infection with *Leishmania infantum*

**DOI:** 10.3390/microorganisms9061214

**Published:** 2021-06-04

**Authors:** Rafael José Vivero, Victor Alfonso Castañeda-Monsalve, Luis Roberto Romero, Gregory D. Hurst, Gloria Cadavid-Restrepo, Claudia Ximena Moreno-Herrera

**Affiliations:** 1Grupo de Microbiodiversidad y Bioprospección, Laboratorio de Biología Celular y Molecular, Universidad Nacional de Colombia sede Medellín, Street 59A #63–20, Medellín 050003, Colombia; vacastanedam@unal.edu.co (V.A.C.-M.); gecadavi@unal.edu.co (G.C.-R.); 2Grupo Investigaciones Biomédicas, Universidad de Sucre, Street 16B #13B-80, Sincelejo, Sucre 700001, Colombia; luisro987@gmail.com; 3Institute of Infection, Veterinary, and Ecological Sciences, Leahurst Campus, University of Liverpool, Neston, Wirral, Liverpool CH64 7TE, UK; g.hurst@liverpool.ac.uk

**Keywords:** *Ralstonia*, *Arsenophonus*, microsporidia, microbiota, *Pintomyia evansi*

## Abstract

*Pintomyia evansi* is recognized by its vectorial competence in the transmission of parasites that cause fatal visceral leishmaniasis in rural and urban environments of the Caribbean coast of Colombia. The effect on and the variation of the gut microbiota in female *P. evansi* infected with *Leishmania infantum* were evaluated under experimental conditions using 16S rRNA Illumina MiSeq sequencing. In the coinfection assay with *L. infantum*, 96.8% of the midgut microbial population was composed mainly of Proteobacteria (71.0%), followed by Cyanobacteria (20.4%), Actinobacteria (2.7%), and Firmicutes (2.7%). In insect controls (uninfected with *L. infantum*) that were treated or not with antibiotics, *Ralstonia* was reported to have high relative abundance (55.1–64.8%), in contrast to guts with a high load of infection from *L. infantum* (23.4–35.9%). ASVs that moderately increased in guts infected with *Leishmania* were *Bacillus* and *Aeromonas*. Kruskal–Wallis nonparametric variance statistical inference showed statistically significant intergroup differences in the guts of *P. evansi* infected and uninfected with *L. infantum* (*p* < 0.05), suggesting that some individuals of the microbiota could induce or restrict *Leishmania* infection. This assay also showed a negative effect of the antibiotic treatment and *L. infantum* infection on the gut microbiota diversity. Endosymbionts, such as Microsporidia infections (<2%), were more often associated with guts without *Leishmania* infection, whereas *Arsenophonus* was only found in guts with a high load of *Leishmania* infection and treated with antibiotics. Finally, this is the first report that showed the potential role of intestinal microbiota in natural populations of *P. evansi* in susceptibility to *L. infantum* infection.

## 1. Introduction

The midgut bacterial genera of phlebotomine sand flies, such as *Acinetobacter*, *Enterobacter*, *Pseudomonas, Bacillus, Serratia, Burkholderia, Erwinia,* and *Pantoea,* among others, have been recorded using microbial and molecular approaches for wild-caught and laboratory-reared populations (Telleira et al., 2018; Omondi and Demir, 2020; Kelly et al., 2017). Environmental factors (temperature, precipitation, humidity, and inorganic toxicants), geographical distribution, development stages, organs, sex, feeding habits, and interactions with endosymbionts (e.g., *Wolbachia*) can also influence the diversity and composition of gut microbiota in sandfly populations [1,2,3].

Several studies with phlebotomines from America have also demonstrated the importance and potential role of intestinal microbiota in susceptibility and competence in developing *Leishmania* infection [2]. Recently, in vivo trials of *Lutzomyia longipalpis* coinfection with different *Leishmania* species and bacterial isolates (*Lysinibacillus, Serratia*, and *Pseudocitrobacter*) reported significant differences, suggesting a powerful effect on inhibiting *Leishmania* survival, and causing the death of promastigotes in a few hours or days [4]. Other studies showed that bacterial richness and diversity progressively decreased in *Leishmania infantum*-infected sand flies as the parasite numbers increased [3]. There exist differences in the microbiota composition based on the distinct physiological stages of the adult insects such as *Lutzomyia intermedia* [5]. The antibiotic-mediated perturbation of the midgut microbiota can also influence sand flies that cannot support parasite growth and metacyclogenesis [3]. These data suggested that the microbiota may determine the ability of disease transmission.

Sand flies can also mount general and specific humoral immune responses (via AMPs and reactive oxygen species) to the presence of both Gram-positive and Gram-negative bacteria [6]. The parasite must compete for space and nutrients with the resident gut bacteria and evade the presence of molecules produced by the bacteria and the digestive enzymes of the host [7,8]. The eco-epidemiological complexity of the vectors in America, protozoan pathogens, and microbiota relationship need to be studied further in depth to identify targets (symbionts or endemic bacteria) that can be used for developing paratransgenic strategies for controlling the sandfly populations and blocking their capacity to transmit leishmaniasis to humans.

To date, no comprehensive studies (tritrophic interactions) on gut microbiota, *Leishmania,* and endosymbionts in natural populations of phlebotomine sand flies from Colombia have been reported. Previous studies have reported *Wolbachia* in wild-caught populations of *Pintomyia evansi* from Colombia [9], which have dominant intestinal bacteria such as *Acinetobacter, Enterobacter*, *Pseudomonas*, *Ochrobactrum* and *Methylobacterium*. These were identified through culture-dependent techniques [10] and with the use of high-throughput sequencing (Illumina MiSeq, Illumina Inc., San Diego, CA, USA) [11]. Due to various conditions listed as easy access, and the recognition of the specific collection area, knowledge of patterns of abundance, and the medical importance of *P. evansi* in the Caribbean ecoregion (vector of visceral leishmaniasis), this species was selected to reinforce, deepen, and understand the dynamics of the gut microbiota in the presence of *L. infantum* infection, in treatment with antibiotics, and with the potential presence of some endosymbionts such as *Wolbachia* and *Cardinium*, among others.

## 2. Materials and Methods

### 2.1. Ethics Statement

Sandfly collection was performed following the parameters of Colombian decree no. 1376 under resolution no. 0207 of 090320 of the Ministry of Environment and Sustainable Development, and permission contract no. 121 of 2016, OTROSí No. 25, for access to genetic resources and its derivative products. Sand flies were collected on private property and permission was received from landowners before sampling.

### 2.2. Study Area and Survey of Sand Flies

Entomological sampling was performed in November 2019, during a period of high rainfall. *P. evansi* was collected in the Villa Paz locality, in the periurban areas of the Municipality of Ovejas, Department of Sucre, Colombia. For this, a Shannon-type light trap was used, located in a peridomiciliary environment, between 18:00 and 22:00. Insects were captured with an oral aspirator to complete 100 individuals per muslin cage (20 × 20 cm). A total of four cages were obtained and placed in expanded polystyrene boxes. The samples were transported alive to the Biomedical Research Laboratory of the University of Sucre, where it was kept at 27–29 °C and with 80–90% relative humidity. The sand flies in the four cages were fed with a sterilized sucrose solution (30%) during transport to the laboratory.

### 2.3. Infections under Experimental Conditions of P. evansi with L. infantum

To evaluate the effect of the gut microbiota of *P. evansi* on its susceptibility to infection with *Leishmania,* two treatments were considered. The first female group of *P. evansi* (*n* = 200) was fed with a 30% sucrose solution, whereas the second female group (*n* = 200) was fed with a sugar solution supplemented with a tetracycline and rifampicin antibiotic cocktail (50 μg/μL; Table 1).

The *P. evansi* group treated only with sugar (control) was not included in this study due to the limitations of the number of individuals collected, and the decision to have a better significant representation of females exposed to *Leishmania* parasites; however, it is also necessary to indicate that we previously reported the gut microbiota core community of *P. evansi* (same location) in this condition in previous research [11].

For the test, defibrillated rabbit blood was used [12]. A strain of *L. infantum* (Trypanosomatid Strain Bank of the Biomedical Research Laboratory, University of Sucre (Sincelejo, Sucre, Colombia) was grown and maintained in RPMI 1640 culture medium (Gibco™). Using a hemocytometer, *L. infantum* promastigotes were counted during the exponential growth phase and added to the decomplemented blood for a final concentration of 5 × 10^6^ parasites/mL.

The infection test was conducted 48 h after the capture of the phlebotomines. The sand flies were depleted of the sugar solution, with and without antibiotics, 24 h before the experiment. Using glass feeders covered with 1- to 3-day-old chicken skin membranes, each group of phlebotomines was supplemented with the decomplemented blood containing the parasites (parasite viability previously checked by microscope at 400× (88). Feeding of the females was performed for 2 h in darkness and under temperature and relative humidity conditions described previously. The feeder was subsequently removed, and a blood sample was again examined under a microscope to verify parasite viability. All procedures were performed following the biosecurity standards for managing potential vector insects. The fed females were kept in muslin cages and supplied with a sucrose solution as a feeding source. On the next postinfection day, the phlebotomine females were separated into non-fed and fed, the latter being kept alive in the laboratory.

### 2.4. Sandfly Washing Procedure and Midgut Dissection

To determine *Leishmania* infection, fed females were dissected 5 days after infection [13,14]. Phlebotomines were sacrificed in 2% Extran^®^ (Thomas Scientific, Swedesboro, NJ, USA) detergent solution for 30 s and subsequently placed in a 1× phosphate-buffered saline (PBS) solution Gibco™ (Waltham, MA, USA) to keep the tissues hydrated. The dissection process of each female was performed in a drop of 1× PBS deposited on a slide plate, under a Carl Zeiss™ stereomicroscope (Oberkochen, Germany). Aided with micro stilettos, the head was removed from the thorax. Then, the digestive tract was removed, dragging the last three segments of the abdomen while holding the rest of the body. Later, the insects were examined under a Carl Zeiss™ (Oberkochen, Germany) microscope at 400× for intestinal parasite search. According to parasite detection and load, the phlebotomines were categorized and grouped as (1) uninfected, (2) insects with low parasite load (1–100 parasites), and (3) with high parasite load (>100 parasites) (Table 1), following the initial protocol of Tesh and Modi, 1984, implemented by Santamaria et al., 2005 [13].

The morphological structures, such as spermathecae, were removed aseptically for classical taxonomic identification. The rest of the body was grouped and used to detect *Wolbachia* and *Cardinium*, as described previously [15,16]. During dissection, parasite visualization, and taxonomic confirmation, some females of *Lutzomyia gomezi* individuals were also found incidentally and included in this study as a control to compare the intestinal microbiota variations between sandfly species. *L. gomezi* can also transmit cutaneous leishmaniasis [17], hence a group of guts was included in the study.

### 2.5. DNA Extraction of Guts from P. evansi

DNA was obtained from groups of *P. evansi* using the ZR Tissues & Insect DNA miniPrep (Zymo Research, Irvine, CA, USA) extraction kit, according to the manufacturer’s instruction, and eluted in a total volume of 100 µL. DNA quantification was performed on a Nanodrop 2000 (Thermo Fisher Scientific, Santa Clara, CA, USA; Table 1). The amplification potential of the 16S rDNA was tested via polymerase chain reaction (PCR) using the primers 27F and 1492R [11]. After confirmation, DNA was dried and sent to sequencing services, where the previous purification step was performed before library construction.

### 2.6. Bacterial 16S rRNA Gene Fragment PCR Amplification and Sequencing

PCR amplicon libraries of the 16S rDNA V4 region were prepared using total DNA as a template, according to the protocol described by the EMP 16S Illumina Amplicon Protocol (https://www.earthmicrobiome.org/protocols-and-standards/16s/ accessed on 15 December 2020, Earth Microbiome Project, 2017) [18], using the primers 515F and 806R with a PrimeStar HS DNA Polymerase (Takara, Kusatsu, Japan) cocktail mix, according to the manufacturer’s instructions [19]. The first PCR round consisted of an initial step of 5 min at 95 °C, followed by 20 cycles of denaturation at 98 °C for 10 s, annealing at 50 °C for 15 s, and extension at 72 °C for 45 s, and a final step at 4 °C until further processing. From these reactions, 1 µL was transferred to fresh PCR cocktail mixes, with each sample containing a corresponding and overlapping primer with two distinct indices and Illumina adapters [20], and subsequently run for 10 cycles using the same temperature and times of the first round. The presence and expected size of PCR products were assessed by agarose gel electrophoresis. The amplified products were purified, normalized, and pooled using the SequalPrep Normalization Plate (Thermo Fisher Scientific, Santa Clara, CA, USA) and subjected to 250 bp paired-end Illumina MiSeq (Illumina Inc., San Diego, CA, USA) sequencing.

### 2.7. Bioinformatics and Statistical Analysis of the Microbiota Data

For the demultiplexed 16S amplicon raw pair-end sequence datasets from each sample, the DADA2 software package (https://github.com/benjjneb/dada2 accessed on 15 December 2020) was used following a sequential pipeline for filtering, denoizing, chimeras, and merging [21]. This resulted in assembled datasets for each amplicon, with trimmed reads based on quality, cut out of the primer sequences, deletion of potentially chimeric sequences to detect the counts of each unique amplicon sequence variant (ASV) across all samples, and their classification using the RDP Naive Bayesian Classifier with taxonomic reference Silva database release 132 (https://www.arb-silva.de/documentation/release-132/ accessed on 15 December 2020) [22]. α- and β-Diversity metrics were estimated using the phyloseq software package (https://joey711.github.io/phyloseq/ accessed on 15 December 2020) [23]. Likewise, a β-diversity analysis of microbial communities associated with the established groups of guts of *P. evansi* infected with *Leishmania* was performed, using a Principal Coordinate Analysis (PCoA) of Bray–Curtis dissimilarities of 16S rRNA data, filtered as ASVs at the genus level. The plot was obtained in the PAST 4.0 package. Also were used a heat map to analyze the beta diversity in the Microbiome Analyst (https://www.microbiomeanalyst.ca (accessed on 15 December 2020)).

## 3. Results

### 3.1. P. evansi Gut Microbiota Composition

The total reads, the number of ASVs, and the five most abundant phylum-family-genus from 16S rRNA gene amplicon sequencing were obtained from 11 samples of *P. evansi* guts, untreated and treated with an antibiotic cocktail (based on quality and taxonomy classification; Table 2). In the coinfection assay, 99.3% of the microbial population was composed of Proteobacteria (71.0%), Cyanobacteria (20.4%), Actinobacteria (2.7%), Firmicutes (2.7%), Bacteroidetes (1.8%), and Acidobacteria (1.0%; Figure 1).

### 3.2. Influence of Leishmania and Antibiotic Cocktail on Gut Microbiota

Gut pools uninfected with *L. infantum* (treated or untreated with antibiotics) reported the *Ralstonia* genera to have high relative abundance (55.1–64.8%; Figure 2a), as opposed to groups with a high load of *L. infantum* infection (23.4–35.9%). ASVs that moderately increased in guts infected with *Leishmania* were *Bacillus* (29.3–36%) (Figure 2a). *Burkholderia, Corynebacterium, Aeromonas and Staphylococcus* have an abundance similar to most samples of *Pi. evansi* (Figure 2a). *L. gomezi* with a high *L. infantum* infection rate presented a similar ASV profile to that of *P. evansi* guts uninfected with *Leishmania* (Figure 2a). The sample of *Pi. evansi* only fed with sugar has a microbiota more diverse before of the exposition to *Leishmania* or antibiotics (Figure 2a).

### 3.3. Endosymbiont Detection

No ASVs of *Wolbachia*, *Cardinium*, *Rickettsia,* and *Flavobacterium* were found in the groups of *P. evansi*. The rest of the body (head, thorax, legs, tegument, and reproductive structures) of *P. evansi* females treated and untreated with antibiotics was also identified as negative for *Wolbachia* by conventional PCR. However, other endosymbionts, such as Microsporidia (<2%), were detected in midgut groups without *Leishmania* infection; in groups treated (10-A*Le*^−^ 0.17%; ASVs = 1867 and 12-A*Le*^−^ 0.02%; ASVs = 20) and untreated with an antibiotic mixture (*4Le*^−^ 0.4%; ASVs = 258) containing a low load of *Leishmania* infection (9b-A*Le^+^* 1.9%; ASVs = 542); and in a single group of intestines with a high load of *Leishmania* infection treated with the antibiotic cocktail (7-A*Le^+^* 0.3%; ASVs = 281). Additionally, *Arsenophonus* was found in a gut group with a high load of *Leishmania* infection treated with antibiotics (8-A*Le^+^* 4%; ASVs = 2239).

### 3.4. Diversity of Gut Microbiota in P. evansi

To analyze α-diversity, most of the intestinal groups treated (with and without antibiotics) and subjected to *Leishmania* infection showed indices of richness (observed Chao 1), diversity (Shannon and Simpson), and dominance of ASVs represented by values not exceeding 50, 2, and 0.7, respectively (Figure 2b), which are considered low compared to natural populations of *P. evansi* unexposed to these treatments or insectary conditions. This result suggested an antibiotic perturbation on the microbiota diversity and the potential specialization or dominance of some communities. However, a different expression of diversity profiles was presented in two gut groups with a high infection load (7-A*Le^+^*_High) and without *Leishmania* infection (10-A*Le*^−^_Uninfected), both treated with antibiotics with values higher than those exposed above.

β-Diversity, measured by phyloseq software package version 3.8 (Bray–Curtis distance matrix) and Past package version 4.04 (Plot of Principal Coordinate Analysis-PCoA), suggested differences in the structure and composition of ASVs between the uninfected gut group and *L. infantum* guts infected with a high and low load (Figure 2c,d). However, at the intergroup level, a sample of intestines (7-A*Le^+^*_High) with a high rate of *Leishmania* infection resulted in a divergent ASV composition (Figure 2c). Kruskal–Wallis nonparametric variance statistical inference and heatmap showed statistically significant intergroup differences in guts of *P. evansi* infected and uninfected with *L. infantum* (<0.05), suggesting the impact of microbiota that may induce or restrict *Leishmania* infection in natural populations (Figure 3). Finally, the *L. gomezi* group with a high rate of *Leishmania* infection showed statistically significant differences in the Mann–Whitney paired test (*p* < 0.005), as well as in the sample 7-A*Le^+^*_High (*p* < 0.0005), as opposed to the rest of the groups of the bioassay. The intragroup analysis also showed statistically significant differences between the ASV communities (*p* < 0.0012), which is demonstrated by the distances in the PCoA (Figure 2d) and the heatmap (Figure 3).

## 4. Discussion

This study contributes information on the influence and potential role of the gut microbiota on the experimental establishment of *L. infantum* in a natural population of *P. evansi,* achieved through sequencing strategies and solid bioinformatics that analyzed richness and microbial diversity. Experiments were also performed to explore the behavior of some specific endosymbionts.

First, this study suggested that a fraction of the intestinal microbiota of *P. evansi* females may have a protective role and/or prevent the development or establishment of *L. infantum*. This hypothesis may be associated with the high parasite infection load in the group of guts treated with the antibiotic cocktail. The microbiota also has a fundamental role in the induction, maturation, and function of the host immune system, modulating host protection from pathogens and infectious diseases [3,24]. Bacteria may also directly inhibit pathogen development, either by hindering the necessary interactions between the pathogen and vector epithelium or through the production of antiparasitic molecules [25], such as antimicrobial peptides (e.g., defensins) [26] and pigment (prodigiosin), mainly through Gram-negative bacteria [27].

Treatment with antibiotics decreased the richness and diversity of gut microbiota on *P. evansi,* but the *Leishmania* infection increased. As in other studies, these findings strengthened the theory that any manipulation that reduces the size and/or diversity of the natural microbiota should enhance the ability of *Leishmania* to establish infections in sand flies or other pathogens in mosquitoes [28]. However, in *Phlebotomus dubosqui*, treatment with the antibiotics results in females highly refractory to the development of transmissible infections [29]. The capacity of the gut bacterial symbionts or resistant microbiota to generate appropriate nutrient stress and osmotic conditions is required for promastigote differentiation and survival [29,30], suggesting that significant differences occur if sandfly populations are in the laboratory or wild.

For sand flies, the few studies that have addressed this relationship also support the role of natural microbiota in inhibiting parasite development [31]. *Ralstonia* directly impacted the establishment of *L. infantum* in *P. evansi*. Its relative abundance was high (65%) in groups of guts uninfected or infected with a low load of *Leishmania* infection. However, the ASVs that moderately increased in guts infected with a high load of *Leishmania,* such as *Bacillus* and *Aeromonas* (Figure 2a), frequently isolated bacteria in sand flies [1,3,10,11]. Defined as an ASV similar to *Ralstonia solanacearum* at the species level, these can be acquired by adult or larval-stage sand flies from several sources. *R. solanacearum* is a Gram-negative and plant pathogenic bacterium [32]. The pathogenic lifestyles of this bacterium are attributed to ecological adaptation and genomic convergence during vertical evolution [32]. The frequency, density, and diversity of phylotypes of *R. solanacearum* in insects are poorly documented, but there are recent reports on *Illeis* [33]*, Oscinella, Aphis mellifera, Chelisoches,* and *Dolichiderus* associated with crops, which postulated that these species could transport *R. solanacearum* [34]. Natural populations of *L. longipalpis*, *Lutzomyia cruzi,* and *L. intermedia* were found in sand flies from Brazil and Colombia [5,35,36]. Further studies are needed to understand the mechanism of interaction between *Leishmania* and *Ralstonia.*

Other ASVs were found in the gut microbiota of *P**. evansi*, with a significant abundance of *Staphylococcus*, *Corynebacterium,* and *Burkholderia*. The first two were previously detected in *Phlebotomus papatasi*, *Phlebotomus argentipes*, *Phlebotomus perniciosus*, and *L.*
*longipalpis* [37,38,39,40], whereas the others have not been registered for sand flies. To date, no information exists on the role of these bacteria in the interaction between *Leishmania* and sand flies.

Second, this study also showed that the endosymbionts (*Wolbachia* and *Cardinium*) were not found by either conventional PCR or next-generation sequencing. Instead, they were positively related to a natural variation in the frequency of infection, to abundance, or to the seasonality of these endosymbionts concerning the behavior of insects. This study did not attribute the absence of these two endosymbionts to the treatment with the mixture of antibiotics supplied to females of *P. evansi*. However, an important finding was the presence of Microsporidia and *Arsenophonus* in the intestinal microbiota of *P. evansi*. The first was more frequent in groups of intestines with no or a low load of the *Leishmania* infection.

This result was interesting because of the previously reported impact of Microsporidia on parasites, such as *Plasmodium* [41], suggesting the potential influence of this endosymbiont on *Leishmania.* Unlike *Arsenophonus* that was only detected in a group of intestines with a high load of *Leishmania* infection, this was the first record of *Arsenophonus* in sand flies from America. *Arsenophonus* has been described to significantly contribute to virus transmission in plants [42,43,44] and has been identified in parasitic wasps, triatomine bugs, psyllids, whiteflies, aphids, ticks, ant lions, hippoboscids, streblids, bees, lice, bat flies, louse flies, and two plant species [45]. The manipulation of host reproduction has been demonstrated by *Arsenophonus* [46], but some strains isolated from a divergent range of arthropods showed no evidence of sex ratio distortion activity [47]. *Arsenophonus* can be easily established in triatomines under laboratory conditions and influence the modification of intestinal microbiota over time and vector competition [48].

Finally, during taxonomic identification in the process of intestinal dissection, we found that a group of infected guts with *Leishmania* corresponded to a female of *L. gomezi*. Because of the importance of *L. gomezi* (endophilic/anthropophilic species) as a vector of the cutaneous leishmaniasis in Colombia [49], principally associated with highly intervened areas on the Caribbean coast, we consider its inclusion, highlighting the surprise experimental infection and first report with *L. infantum*. Despite the potential significance of *L. gomezi* as a vector of *L. panamensis* and *L. braziliensis* [50], few types of research have aimed to describe the associated microbiota under states of interaction with *Leishmania*. We found a microbiota profile consisting mainly of *Ralstonia* ASVs, followed by *Burkholderia-Caballeronia-Paraburkholderia* and ASVs without taxonomic assignment (NA) in a significant percentage. Complementary studies should be carried out to increase the information on the gut microbiota of this vector.

## 5. Conclusions

The inclusion of a larger number of groups of guts of *P. evansi* uninfected with *Leishmania* may improve the analysis. However, this study was done with natural and wild populations, so their abundance and susceptibility to infection are subject to variations. In summary, it is the first study that showed the potential role of the gut microbiota in natural populations of *P. evansi* on their susceptibility to *L. infantum* infection. This study also showed that treatment with antibiotics reduces the richness and diversity of microbiota*,* but *Leishmania* infection increases, indicating that the microbiota can be a barrier to the establishment and development of promastigotes in *P. evansi.* Finally, in vivo coinfection studies are needed to better understand *Leishmania*–microbiota–sand fly interactions and identify microbiome communities or effector molecules determinant in blocking or reducing the development and establishment of *Leishmania.*

## Figures and Tables

**Figure 1 microorganisms-09-01214-f001:**
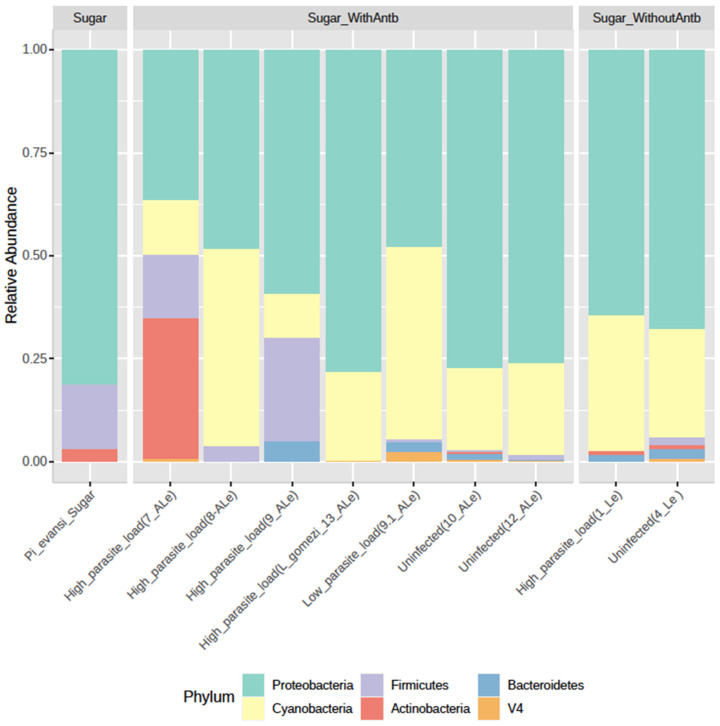
Gut microbiota composition at the phylum level in wild specimens of several natural populations of *P. evansi* infected with *Leishmania* and treated with antibiotics. The relative abundance of ASVs that were called to the taxonomic rank of the phylum.

**Figure 2 microorganisms-09-01214-f002:**
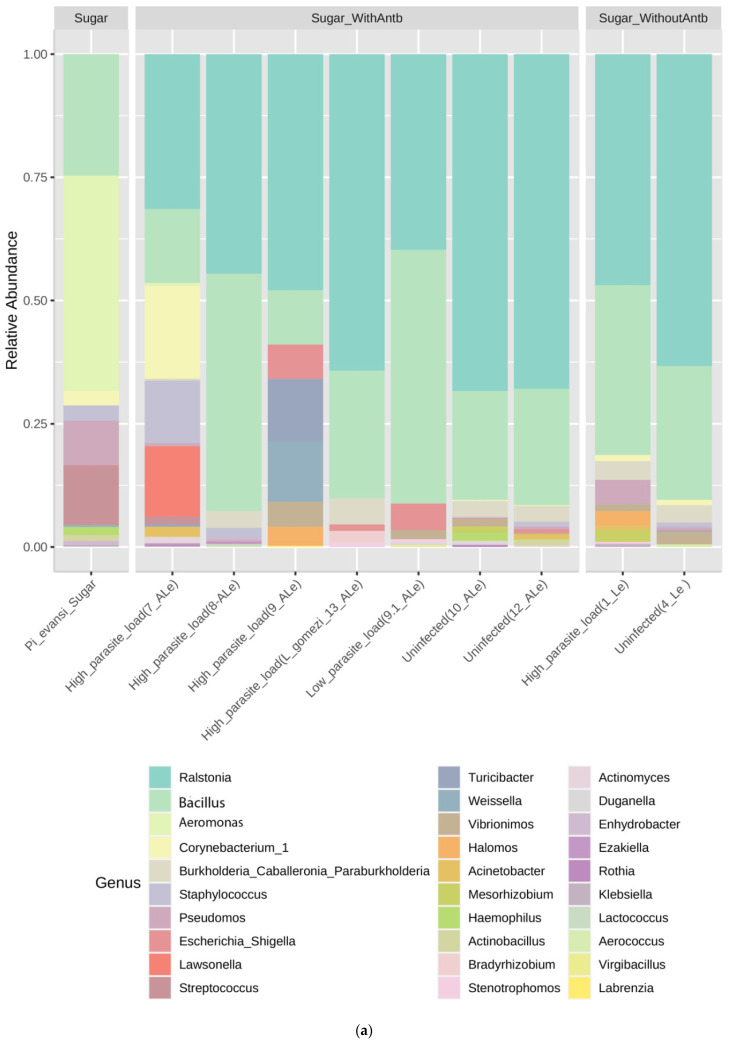
Gut microbiota composition and diversity in wild specimens of *P. evansi* from the north of Colombia, infected with *Leishmania* and treated with antibiotics. (**a**) Relative abundance of ASVs that were called to the taxonomic rank of genus. Taxa with <0.5% relative abundance were grouped together as “Genus < 0.5%”. (**b**) α-Diversity index of ASVs of guts of *P. evansi* infected with *Leishmania* and treated with antibiotics. (**c**) Hierarchical clustering analysis (β-diversity) of ASVs at the genus level (**d**) β-diversity analysis of microbial communities associated with the established groups of guts of *P. evansi* infected with *Leishmania*, using a Principal Coordinate Analysis (PCoA) of Bray–Curtis dissimilarities of 16S rRNA data, filtered as ASVs at the genus level. See Table 1 for the detailed nomenclature of *P. evansi* gut pools*. Le^+^* High, guts with high load of *Leishmania* and without antibiotics; *Le^-^* Uninfected, guts uninfected and without antibiotics; A*Le^+^* High-Low, guts with a high or low load of *Leishmania* and treated with antibiotics; A*Le*^−^ Uninfected, guts uninfected but treated with antibiotics.

**Figure 3 microorganisms-09-01214-f003:**
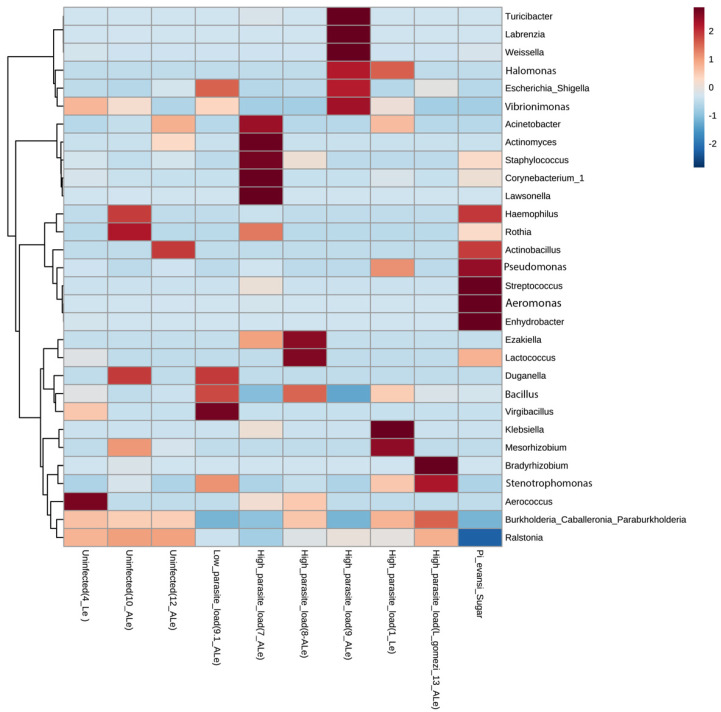
Heatmap based on microbiota composition at genus level associated with gut microbiota in wild specimens of *P. evansi* from the north of Colombia, infected with *Leishmania* and treated with antibiotics. Hierarchical Ward’s linkage clustering based on the Pearson’s correlation coefficient of the microbial taxa abundance. Blue and red colors represent positive and negative correlations, respectively. The color scale represents the scaled abundance of each variable, denoted as Z-score, with red indicating high abundance, and blue indicating low abundance.

**Table 1 microorganisms-09-01214-t001:** Treatments and parasite load of *L. infantum* in gut groups of fed female *P. evansi* and *L. gomezi* from Ovejas, Sucre Department, Colombia.

Treatments	Sandfly	Group Code	No. Guts per Groups	Parasite Load Observed *	DNA Total Concentration (ng/μL)
Sugar solution (30%) supplemented with an antibiotic cocktail (50 μg/μL) and *L. infantum* (5 × 10^6^ parasites/mL)	*P. evansi*	7-A*Le^+^*	15	High	22.1
		8-A*Le^+^*	15	High	16.3
		9-A*Le^+^*	12	High	17.5
		9.1-A*Le^+^*	7	Low	23.8
		10-A*Le*^−^	10	Uninfected	15.1
		12-A*Le*^−^	8	Uninfected	20.3
Sugar solution (30%) and *L. infantum* (5 × 10^6^ parasites/mL)		1-*Le^+^*	5	High	19.7
		4-*Le*^−^	3	Uninfected	19.8
Sugar solution (30%) supplemented with an antibiotic cocktail (50 μg/μL) and *L. infantum* (5 × 10^6^ parasites/mL)	*L. gomezi*	13-A*Le^+^*	2	High	13.8

***** Parasite load observed. Uninfected (0), low (1–100), and high (>100) (Romero-Ricardo, data unpublished).

**Table 2 microorganisms-09-01214-t002:** Summary of the results obtained from 16S rRNA gene amplicon sequencing of *P. evansi* gut microbiota under experimental infection with *L. infantum* and treated with antibiotics.

	*P. evansi* Coinfection	*P. evansi* Coinfection
	Dataset Untreated	Dataset Treated
**Total Reads**	1′551.612	1′540.250
**No. ASVs**	415	227
**Phyla**	(14)	(11)
	Acidobacteria	Acidobacteria
	Actinobacteria	Actinobacteria
	Armatimonadetes	Armatimonadetes
	Bacteroidetes	Bacteroidetes
	Cyanobacteria	Cyanobacteria
	Deinococcus–Thermus	Firmicutes
	Euglenozoa	Fusobacteria
	Firmicutes	Microsporidia
	Fusobacteria	Proteobacteria
	Microsporidia	Tenericutes
	Patescibacteria	Verrucomicrobia
	Proteobacteria	
	Tenericutes	
	Verrucomicrobia	
**5 major families (total counts)**	Burkholderiaceae	Burkholderiaceae
	Bacillaceae	Bacillaceae
	Corynebacteriaceae	Corynebacteriaceae
	Chitinophagaceae	Chitinophagaceae
	Elsteraceae	Elsteraceae
**Top 6 most abundant bacterial** **genera (total counts)**	*Ralstonia*	*Ralstonia*
	Bacillus	Bacillus
	*Burkholderia-Caballeronia-Paraburkholderia*	*Burkholderia-Caballeronia-Paraburkholderia*
	*Vibrionimonas*	*Vibrionimonas*
	*Corynebacterium* *Staphylococcus*	*Corynebacterium* *Staphylococcus*
**No. taxa summarized to** **the genus level**	141	87

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
