# Peer review of "Gut Microbiota Dynamics in Natural Populations of Pintomyia evansi under Experimental Infection with Leishmania infantum"

_microorganisms, 2021, doi:10.3390/microorganisms9061214_

Round 1

Reviewer 1 Report

Studies on the role of phlebotomine gut microbiota on the transmission of Leishmania parasites are still scarce. This study is a valuable contribution for this field of investigation.

I have one major question concerning the study design: the groups of female sand flies investigated were either treated with sugar solutions plus Leishmania parasites plus or without antibiotics. Why there was no control sandfly group in sugar solutions only, without antibiotics or parasites? And why another sand fly species (Lutzomyia gomezi) has been included in the study for which only one group (with antibiotics and high parasite load) was available? The results for this latter group are not further discussed in the paper.

In the tables and figures there is one abbreviation “NA”, what does it mean?

The English needs some polishing throughout the text, it should checked by a native speaker.

Author Response

Gut microbiota dynamics in natural populations of Pintomyia evansi under experimental infection with Leishmania infantum

Microorganisms-1139856

Reviewer 1- Reviewer 2

  1. I have one major question concerning the study design: the groups of female sand flies investigated were either treated with sugar solutions plus Leishmania parasites plus or without antibiotics. Why there was no control sandfly group in sugar solutions only, without antibiotics or parasites? 

Response:

We appreciate the suggestion.

Pi. evansi were initially fed with sugary solution from the collection area to the laboratory. Subsequently, two boxes of phlebotomines were treated with an antibiotic cocktail and two boxes continued their treatment with the sugary solution before exposure or experimental feeding with blood and Leishmania.

The P. evansi group treated only with sugar shows the basal profile of the microbiota and allows to see the change against the exposure of Leishmaniaand antibiotics. However, we consider having a sufficient and significant representation of phlebotomines exposed to Leishmania parasites and seeing the influence or role of the microbiota to favor the establishment of Leishmania. The populations are natural, collected during one or two nights. They are not from the laboratory. We guarantee with the number of females collected to have two groups (treated and not treated with antibiotics). We also consider having a third group (only fed with sugar and not exposed to Leishmania), but we had limited resources for sequencing a certain number of samples. Additionally, this information on groups of Pi. evansi (same wild population) treated only with sugar before dissection of intestines, not exposed to parasites, not exposed to antibiotics was previously analyzed by our group in previous research “Wild specimens of sand fly phlebotomine Pintomyia evansi, vector of leishmaniasis, show high abundance of Methylobacterium and natural carriage of Wolbachiaand Cardinium types in the midgut microbiome” doi: 10.1038/s41598-019-53769-z.

Considering the above and  following the suggestion, we add the following sentences:

Topic 2.2. Study area and survey of sand flies

Line 100-101: Sandflies into the four cages, were fed with a sterilized sucrose solution (30%) during transport to the laboratory.

Topic 2.3. Infections under experimental conditions of Pintomyia evansi with Leishmania infantum

Line 109-113:  The P. evansi group treated only with sugar (control) was not included in this study by limitations on the number of individuals collected, and considering having a better significant representation of females exposed to Leishmania parasites; however, it is also necessary to indicate that we previously report the gut microbiota core community of Pi. evansi (same location) in this condition in previous research (11).

  1. And why another sand fly species (Lutzomyia gomezi) has been included in the study for which only one group (with antibiotics and high parasite load) was available?

Response:

Pintomyia evansi is one of the most abundant species (98%) in the study area and is the main vector of Leishmania infantum transmission in the Caribbean region of Colombia. However, when collecting in the Shanon trap several species may remain in the cages. Because of the complexity of separating these species by morphotypes based on external structures, we found during taxonomic identification in the process of intestinal dissection that a group of infected guts corresponded to a female of Lu. gomezi. Because of the importance of Lu. gomezi as a vector of the cutaneous Leishmaniasis and because it is the second most important species in the Caribbean region, we consider its inclusion to be of interest, especially since it has experimentally infected a species of the parasite such as Leishmania infantum, symbolizing a finding of interest in public health.

Considering the above and  following the suggestion, we add the following sentences:

Topic 4. Discussion

Line 373-384: Finally, we found during taxonomic identification in the process of intestinal dissection a group of infected guts with Leishmania corresponded to a female of L. gomezi. Because of the importance of L. gomezi(endophilic/anthropophilic species) as a vector of the cutaneous leishmaniasis in Colombia [49], principally associated with highly intervened areas on the Caribbean coast, we consider its inclusion, highlighting the surprise experimental infection and first report with L. infantum. Despite the potential significance of L. gomezi as a vector of L. panamensisand L. braziliensis [50], few types of research have targeted to describe the associated microbiota under states of interaction with Leishmania. We found a microbiota profile consisting mainly of Ralstonia ASVs followed by Burkholderia-Caballeronia-Paraburkholderia and ASVs without taxonomic assignment (NA) in a significant percentage. Complementary studies should be carried out to increase the information on the gut microbiota of this vector.

  1. In the tables and figures there is one abbreviation “NA”, what does it mean?

Response:

The information associated with the abbreviation was included in tables and figures.

  1. The English needs some polishing throughout the text, it should checked by a native speaker.

Yes. The manuscript is checked by Company ENAGO (https://www.enago.com) for style revision and a certificate will be attached.

Reviewer 2.

  1. Although the data are promising, supportive data are missing. For example, when the flies are given a mock or antibiotic treatment, there is no precise quantification of parasite burden. It is essential that the authors provide pictures to illustrate differences between these two groups. This is also because antiobiotic treatment may be altering gut physiology and morphology on top of microbiota composition.

Response:  The percentage of infection was established by the traditional method, for which dissection and visualization of the intestines were performed. A photographic record of the morphology of intestines with and without antibiotic treatment is a good suggestion. But we did not have the infrastructure for a quality record of the images of high-resolution and the intestinal contents had to be stored quickly to avoid contamination, form the groups correctly, and have success in the amplification by PCR of 16S. Even do Leishmania PCR is additional support, but the performance and method of obtaining total DNA for microbiota analysis are different and need optimization of all intestinal content for Illumina sequencing.

We accept your suggestion for another study that will evaluate the Microbiota and Leishmania infection versus treatment with antibiotics in another species of phlebotomine.

Considering the above and  following the suggestion, we add the following sentences:

Topic 2.4. Sandfly washing procedure and midgut dissection

Line 145-146:  following the initial protocol of Tesh & Modi 1984, implemented by Santamaria et al., 2005 [13].

  1. The figures in this manuscript are often blurry and not very clear. Higher resolution images (600 DPI is recommended) are needed. The manuscript also suffers from a plethora of linguistic mistakes that often make the text hard to read. Scientific names are often not capitalized in the discussion. Hence, a thorough and conscientious revision is required.

The manuscript is checked by Company ENAGO (https://www.enago.com) for style revision and a certificate will be attached.

The figures are being improved in their resolution for better analysis and visualization

Additional references were included

  1. Duque, P; Vélez, I.; Morales, M.; Sierra,  Sand flies fauna involved in the transmission of cutaneous leishmaniasis in Afro-Colombian and Amerindian communities of Choco, Pacific Coast of Colombia. Neotropical Entomology. 2004. 33(2); 255-264. https://dx.doi.org/10.1590/S1519-566X2004000200018.
  2. Cortés, Alemán,; Pérez-Doria, A.; & Bejarano, E. Flebotomíneos (Diptera: Psychodidae) antropofílicos de importancia en salud pública en Los Montes de María, Colombia.Revista Cubana de Medicina Tropical61(3), 220-225.

Reviewer 2 Report

In this manuscript, Vivero and colleagues address the role of sand fly (P. evansi) microbiota on L. infantum infection. They find that L. infantum infection alters microbiota composition and diversity, and they show that an antibiotic cocktail given to sand flies enhances Leishmania infection. 

Although the data are promising, supportive data are missing. For example, when the flies are given a mock or antibiotic treatment, there is no precise quantification of parasite burden. It is essential that the authors provide pictures to illustrate differences between these two groups. This is also because antiobiotic treatment may be altering gut physiology and morphology on top of microbiota composition. 

The figures in this  manuscript are often blurry and not very clear. Higher resolution images (600 DPI is recommended) are needed. The manuscript also suffers from a plethora of linguistic mistakes that often make the text hard to read. Scientific names are often not capitalized in the discussion. Hence, a thorough and conscientious revision is required.

Author Response

(The authors gave the same response as above.)

Round 2

Reviewer 1 Report

The authors have addressed my previous major question concerning the study design sufficiently.

I have still some minor comments.:

The authors should fully spell abbreviations when they appear the first time in the text., e.g. ASV in the abstract lane 25.

In the table 2A there is still the abbreviation “NA”, which is not explained.

M&M; lane 112: in the newly added text it should be “previously reported”.

Discussion:
Lane 362: it should be “the endosymbionts” instead of “these endosymbionts”!
Lanes 374-376: I do not understand the sentence starting with “Unlike Arsenophonus…..”.
The last paragraph of the discussion (before conclusions) needs some revision of the English. I did not fully understand the message of the authors here.

Reviewer 2 Report

The authors have submitted a textual revision of their original manuscript. For future manuscripts, the authors are advised to improve on the quality of their experimental setups.